# Stakeholder perspectives on interventions to improve HIV pre-exposure prophylaxis uptake and continuation in Lesotho: A participant-ranked preferences study

**Joy J. Chebet**[1], **Shannon A. McMahon**[2,3], **Rachel P. Chase**[4], **Tapiwa Tarumbiswa**[5], **Chivimbiso Maponga**[6], **Esther Mandara**[6], **Till Bärnighausen**[2,7,8], **Pascal Geldsetzer**[9,10]*

1 Department of Health Promotion Sciences, Mel and Enid Zuckerman College of Public Health, University of Arizona, Tucson, AZ, United States of America, 2 Heidelberg Institute of Global Health, Heidelberg University, Germany, 3 Social and Behavioral Interventions, Johns Hopkins Bloomberg School of Public Health, Baltimore, MD, United States of America, 4 Wexner Medical Center, Ohio State University, Columbus, OH, United States of America, 5 Disease Control Department, Ministry of Health Lesotho, Maseru, Lesotho, 6 Clinton Health Access Initiative–Lesotho Country Office, Maseru, Lesotho, 7 Department of Global Health and Population, Harvard T.H. Chan School of Public Health, Boston, Massachusetts, United States of America, 8 Africa Health Research Institute (AHRI), Durban, South Africa, 9 Division of Primary Care and Population Health, Department of Medicine, Stanford University, CA, United States of America, 10 Chan Zuckerberg Biohub–San Francisco, San Francisco, CA, United States of America

* pgeldsetzer@stanford.edu

**Data Availability Statement:** All data can be found in the supporting information files.

## Abstract

Low uptake and high discontinuation remain major obstacles to realizing the potential of Pre-Exposure Prophylaxis (PrEP) in changing the trajectory of the HIV epidemic. We conducted a card sorting and ranking exercise with 155 local stakeholders to determine their views on the most important barriers and most promising interventions to achieving high PrEP coverage. Stakeholders were a purposive sample of PrEP policymakers and implementing partners (n = 7), healthcare providers (n = 51), and end-users (n = 97). End-users included adults who were currently using PrEP (n = 55), formerly using PrEP (n = 36), and those who were offered PrEP but declined (n = 6). Participants sorted pre-selected interventions and barriers to PrEP coverage into three piles–most, somewhat, and least important. Participants then ranked interventions and barriers in the "most important" piles in ascending order of significance. Ranked preferences were analyzed as voting data to identify the smallest set of candidates for which each candidate in the set would win in a two-candidate election against any candidate outside the set. Participants viewed a lack of PrEP awareness as the most important barrier to PrEP uptake for women, and a fear of HIV testing for men. Community-based HIV testing was ranked as the most promising intervention to improve PrEP uptake for both men and women. Perceived or experienced stigma was seen as an important barrier for PrEP continuation for both men and women, with an additional important barrier for men being daily activities that compete with the time needed to take a daily pill. Adherence counseling and multi-month PrEP prescriptions were seen as the most promising interventions to improve PrEP continuation. Our findings suggest community-based activities that generate PrEP demand (community-based HIV testing and mass

**Funding:** This study was supported by the Alexander-von-Humboldt Foundation University Professorship to TB (no grant number; https://www.humboldt-foundation.de/en/). PG is a Chan Zuckerberg Biohub investigator (no grant number; https://www.czbiohub.org). The funders had no role in study design, data collection and analysis, decision to publish, or preparation of the manuscript.

**Competing interests:** The authors have declared that no competing interests exist.

media campaigns), reinforced with facility-based follow-up (counseling and multi-month prescription) could be promising interventions for PrEP programs that are aimed at the general adult population.

## Introduction

Despite declines in infection and transmission rates over the last three decades, stagnating progress towards the goal of ending the HIV epidemic indicates a need to expand effective prevention programs to address current gaps [1]. Interventions focusing solely on behavior change have demonstrated limited success in preventing HIV infection at the population level [2]. Conversely, the use of antiretroviral (ARV) drugs have shown the capacity to acutely address the global HIV burden through a strategy that: 1) identifies those who are HIV positive with the aim of achieving viral load suppression among these patients through consistent use [3, 4], and 2) provides once-daily oral Pre-Exposure Prophylaxis (PrEP) to those at substantial risk of acquiring HIV to prevent infection [5, 6].

Clinical trials and demonstration projects have shown PrEP to be over 90% efficacious in preventing HIV infection when used consistently and as directed [7]. However, the real-life effectiveness of PrEP is strongly dependent on adherence to PrEP [7]. Thus, both low uptake and discontinuation of PrEP remain substantial obstacles to achieving large-scale PrEP coverage [8–10]. Previous studies have found that impediments that dissuade users from initial enrollment and sustained PrEP use include: 1) individual-level barriers, including fear of HIV testing, concern about adverse side effects, perception of low HIV infection risk, and disbelief in the drug's efficacy [11–14]; 2) social-level barriers encompassing concern over communicating about sexual matters with healthcare providers, perceived or experienced stigma surrounding PrEP use, limited decision-making capacity and lack of a robust social support system [11, 13, 15]; and 3) structural barriers including, limited awareness of PrEP and access to PrEP-related services [11–14]. Barriers specific to retention documented in the literature include challenges related to practical difficulties of taking a daily pill and life stressors that compete for time and mental bandwidth [16, 17].

Interventions to improve PrEP uptake have focused on demand creation methods implemented through mass media campaigns and direct promotion in various settings, including the workplace, social gatherings and health facilities [17]. For messaging to reach the grassroots, collaboration and partnerships with local leaders, as well as community- and faith-based organizations have been encouraged [12, 15]. Interventions to improve retention, on the other hand, have included increased contact between the health system and the user (through text messages and phone calls), increased adherence support (achieved through extended facility hours, adherence counseling, and support groups), incentivizing PrEP use, and providing multi-month prescriptions to reduce the burden placed on users [16, 17].

Current evidence on barriers and interventions to improve PrEP coverage has mostly drawn on information from specific population groups classified as being at substantial risk for HIV infection [18]. This has included Men who have Sex with Men (MSM), Injection Drug Users (IDUs), serodiscordant couples, and adolescent girls and women [19–24]. While recent studies have contributed evidence on specific interventions to improve PrEP uptake more broadly–such as counseling [25] and integration of PrEP services into HIV clinics [26]–more data on other promising interventions to achieve high PrEP coverage in the general adult population are needed, particularly in the sub-Saharan Africa region. Using a novel participatory

card sorting and ranking methodology and by studying the views of a broad set of stakeholders in Lesotho, the objective of this study was to inform PrEP implementation efforts for the general population. Specifically, this study aimed to determine stakeholders' views on which are 1) the most important reasons for low uptake and discontinuation of PrEP, and 2) the most promising interventions for improving PrEP uptake and continuation.

## Methods

### Study setting and selection of study sites

The Lesotho Ministry of Health began offering PrEP as part of a comprehensive HIV prevention package in 2016, with the program focusing largely on serodiscordant couples [27]. This approach was employed to address the generalized HIV epidemic in the country–the second highest global prevalence of HIV among adults aged 15–59 years (25.6%) [28]. Since the PrEP program's initiation in a subset of the country's ten districts, it has been expanded to include all individuals at substantial risk of HIV infection [27]. This is defined as populations with an HIV incidence rate of 3 per 100 person-years on a population level [27, 29]. Currently, the Ministry of Health is scaling up the PrEP program to reach all 10 districts in the country [27].

Our study was conducted in five districts where the PrEP program was initially implemented: Maseru, Leribe, Berea, Mafeteng and Mohales Hoek. Two healthcare facilities were identified as study sites in each district. To capture variation in setting, the study sites were purposively selected to include a range of PrEP client volumes, governmental versus private facilities, and rural versus urban areas.

### Study design and sampling

Data were collected between March and April 2019 as part of a larger implementation research study on stakeholder perspectives on the Lesotho PrEP program and scale-up [30]. While other sub-studies aimed to explore participant knowledge of PrEP and motivations for use, the present study sought to understand stakeholder perspectives on which barriers to PrEP uptake and interventions encourage its use with the ultimate aim of identifying salient interventions for further investigation in future. The use of social choice theory, and the Smith set/Condorcet method specifically, allowed us to reach group consensus and identify robust interventions to be considered to improve PrEP coverage [31, 32]. These techniques allow for nuanced participant-centered insights on priority barriers/interventions and integrates varied perspectives from PrEP stakeholders involved in implementation and use of PrEP. Given the broad nature of our inquiry, other preference elicitation methods, such as discrete choice experiments, were inappropriate [33].

Participants were identified based on their engagement with the national PrEP program and represented multiple levels of the health system–from policymakers to target end-users. Participants were purposively selected to participate in the study based on their expertise and experience with PrEP policy development, implementation, and/or PrEP use. The sampling method aimed to provide variation in perspective and viewpoints. The stakeholders along with their inclusion criteria and sampling strategy are summarized in Table 1.

### Card sorting and ranking exercise

Barriers and interventions included in this study were sourced *a priori* from literature [11–17] and through discussions with collaborators in the PrEP field. Following the ecological model, we selected barriers to represent all levels of the health system–individual, interpersonal, societal, contextual and structural factors. The barriers and interventions to PrEP implementation

**Table 1. Summary of participant sampling strategies and inclusion criteria.**

| Participant group (N = 155) | Inclusion criteria | Sampling strategy |
|---|---|---|
| Policymakers and implementing partners (n = 7) | Expertise and experience with developing PrEP policy, overseeing PrEP programs, and/or implementing PrEP programs. | Identified in collaboration with the Ministry of Health and purposively selected to represent expertise and experience in PrEP policy development and program implementation. |
| Healthcare providers (n = 51) | Experience in providing PrEP services directly to clients. | Identified based on inclusion criterion at study sites; purposively selected to represent diversity in years of HIV experience, gender and cadre. |
| Current PrEP users (n = 55) | Individuals actively using PrEP at the time of the interview, regardless of duration of use and/or previous interrupted use. | Identified through facility records and purposively selected to represent diversity in age, gender, educational attainment and duration of PrEP use. |
| Former PrEP users (n = 36) | Individuals who had at one time used PrEP, but at the time of the interview were not on the drug, regardless of duration of non-use. | Identified through facility records and Community Based Organizations (CBOs) working with populations at high risk of HIV infection. Participants were purposively selected to represent diversity in age, gender, educational attainment and duration of PrEP use prior to discontinuation. |
| PrEP decliners (n = 6) | Individuals who were encouraged to initiate PrEP following consultation with a health provider who determined them to be at high risk for HIV infection, but declined to use the drug. | Identified through facility records and CBOs working with populations at high risk of HIV infection. Participants were purposively selected to represent diversity in age, gender and educational attainment. |

that were presented to participants in this study are shown in supplementary material S1 Table. These barrier and intervention candidates were written on individual cards and presented to each participant during a one-on-one interview.

The card sorting and ranking exercise had two steps (Fig 1). First, the sorting portion of the exercise sought to identify the most important barriers and most helpful interventions for the

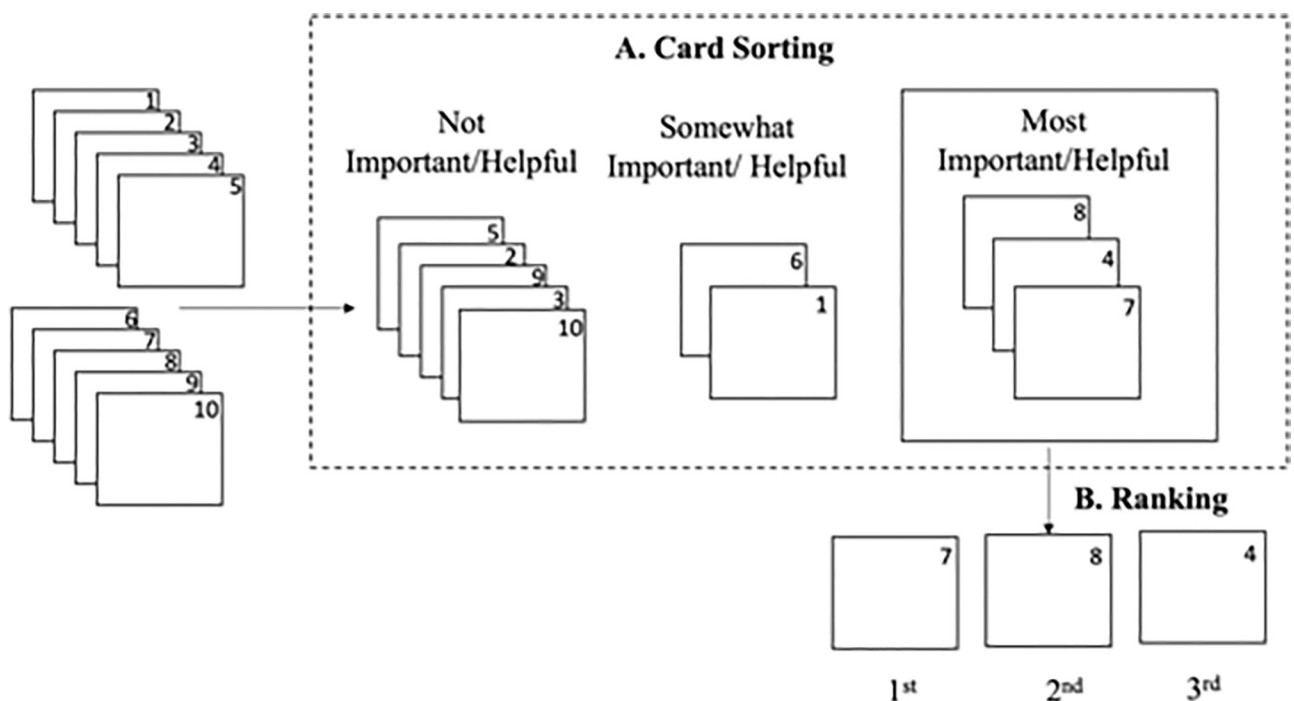

**Fig 1. Schematic illustrating hypothetical card sorting into piles based on importance, and ranking of candidates in the 'most important/helpful' pile in ascending order.**

uptake and retention of PrEP for men and women in Lesotho from the participant's perspective. Participants were asked to place barriers and interventions into three piles with pre-determined themes. For barriers, the piles were: 1) most important; 2) somewhat important; and 3) not important. Intervention piles were: 1) most helpful; 2) somewhat helpful; and 3) not helpful. To overcome literacy challenges and to ensure consistent interpretation across participants, research assistants presented each of the candidates on separate laminated cards and verbally explained the barrier/intervention through moderated facilitation. Second, in order to prioritize the most important barriers and most helpful interventions, participants ranked candidates under the "most important barrier" and "most helpful intervention" piles in ascending order, with the candidate ranked first being the most important barrier or most helpful intervention. Participants were prohibited from placing two or more barriers/interventions in the same position–that is, no ties were allowed. To gain gender-specific insight, each participant (regardless of their gender) was asked to sort and rank each set of barriers and interventions for men and women separately. The instrument used to carry out the pile sorting and ranking exercise is available as supplemental information S1 Text.

## Data management and quality control

Research assistants recorded each participant's preferences physically on a paper guide, reviewed the data collected following each interview and asked for clarification from participants before their departure. A study investigator (JJC) then reviewed all the research assistant questionnaires for accuracy and completeness. Participant responses were then entered into a Microsoft Excel spreadsheet. Two investigators (JJC and RPC) independently reviewed entered data for inaccuracies, such as ties in ranking and other entry errors (see S1 Data). Discrepancies were resolved by cross referencing with the physical questionnaires.

## Data analysis

Ranked preferences were treated like voting data, whereby each choice presented to a participant was considered a candidate running in an election. Candidates sorted into the "most important barrier" or "most helpful intervention" pile were pitted against each other in pairwise, head-to-head contests, with the participant's vote going to the candidate they ranked higher. We identified the Smith set, which is the smallest set of candidates wherein each member in the set would win in a head-to-head election against any candidate outside of the set [31]. The candidates in the Smith set would therefore be considered to have mutual majority. When there is only one candidate in the Smith set, this candidate is the Condorcet winner [32]. The Condorcet winner, thus, is a candidate that would win against all the other candidates in a head-to-head election [32]. In line with our objective of identifying the most highly prioritized barriers/interventions for PrEP coverage, a single winner was not essential. Rather, we aimed to identify highly prioritized interventions for PrEP coverage for further evaluation for financial, cultural and practical feasibility in improving PrEP coverage.

## Ethics statement

Ethical approval for this study was received from the research and ethics committee of the Lesotho Ministry of Health (ID03-2019), and the Heidelberg University ethical review board (S-865/2018). All participants provided written informed consent prior to participation in the study.

## Results

### Participant characteristics

We enrolled 155 participants in the study. Policymakers (n = 4) included one participant involved in the oversight of the PrEP program at the national level, and three participants responsible for the dissemination and implementation of the PrEP program at the district level. On average, policymakers participating in the study had been in their current position for 6.5 years (range: 3–12 years) at the time of the interview. Implementing partners (n = 3) included advisors and managers whose organizations were directly involved in the development of HIV-related policies and implementation of the national PrEP program. On average, the participating implementing partners had been in their current position for 2.7 years (range: 2–3 years). Due to their small number and similarities, policymakers and implementing partners were grouped for analyses. The demographic characteristics of all participants are detailed in Table 2. Note that while 105 health providers were included in the larger implementation study, 51 of them participated in the card sorting and ranking exercise. However, data from this subset were prematurely anonymized. Therefore, health provider demographic information from the total sample of 105 health providers is used to approximate this study's health provider sample.

### Barriers to PrEP uptake

When asked to sort and rank barriers related to PrEP uptake in Lesotho, participant prioritizations revealed differences in perceptions of obstacles that hinder men and women from initiating PrEP (see Fig 2A). For men, overwhelmingly and across all respondent groups, participants ranked fear of HIV testing as the biggest barrier to initiating PrEP. For women, this was not prioritized as a prominent barrier. Instead, lack of awareness was prioritized as a substantial barrier for PrEP initiation for women. Among social-related barriers for PrEP initiation, discussing sexual matters with healthcare providers and perceived stigma were prioritized as a more important barrier for men than women. However, other socially relevant barriers, including lack of social support and limited decision-making power were more highly prioritized for women than men. Of note, concerns of side effects and the perception that PrEP is not effective were not prioritized as important barriers for uptake for neither men nor women.

In head-to-head pairwise elections (Table 3), fear of HIV testing emerged as the most important barrier to PrEP uptake for men. This candidate persisted as the winner in disaggregated data among all respondent groups, with the exception of PrEP decliners, for whom both fear of HIV testing and lack of awareness emerged as winners (as a Smith set). For women, lack of awareness was the winning candidate as biggest barrier to uptake. This barrier remained the winning candidate among all respondent groups, except among policymakers/implementing partners and healthcare providers. Lack of awareness and perceived stigma were the winners according to policymakers/implementing partners. Among healthcare providers, the winners were limited awareness, perceived sigma, and lack of social support.

### Interventions to improve PrEP uptake

There were no substantial differences between the most highly participant-prioritized interventions for PrEP uptake by target population gender (see Fig 2B). Participants ranked community-based HIV testing as the most helpful intervention for PrEP uptake for both men and women. Similarly, mass media campaigns were highly prioritized for both genders. However, there were some gendered differences in the less prioritized PrEP uptake interventions.

**Table 2. Demographic characteristics of participants.**

| | | Policymakers and Implementing Partners (n = 7) | Healthcare Providers[1] (n = 51) | Current Users (n = 55) | Former Users (n = 36) | Decliners (n = 6) |
|---|---|---|---|---|---|---|
| **Age**: years; mean (sd); range | | 47.4 (5.2); 41–56 | 37.3 (11.2); 20–65 | 36.4 (12.6); 20–71 | 26.7 (9.7); 18–62 | 28.8 (5.1); 22–34 |
| **Female**: n (%) | | 5 (71.4) | 82 (78.1) | 38 (69.1) | 33 (91.7) | 6 (100) |
| **District**: n (%) | Maseru | 4 (57.1) | 24 (22.9) | 7 (12.7) | 31 (86.1) | 4 (66.7) |
| | Leribe | 1 (14.3) | 16 (15.2) | 12 (21.8) | 2 (5.6) | 1 (16.7) |
| | Berea | 0 | 33 (31.4) | 9 (16.4) | 1 (2.8) | 0 |
| | Mafeteng | 1 (14.3) | 18 (17.1) | 20 (36.4) | 2 (5.6) | 1 (16.7) |
| | Mohales Hoek | 1 (14.3) | 14 (13.3) | 7 (12.7) | 0 | 0 |
| **Urban interview location**: n (%) | | 7 (100) | 52 (49.5) | 32 (58.2) | 36 (100) | 5 (83.3) |
| **Years in position**[2]: mean (sd); range | | 4.9 (3.5); 2–12 | 6.2 (6.0) | N/A | N/A | N/A |
| **Educational attainment**[3]: n (%) | None or some primary | 0 | - | 18 (32.7) | 4 (11.4) | 0 |
| | Completed primary school | 0 | - | 3 (5.5) | 3 (8.6) | 0 |
| | Some high school | 0 | - | 17 (30.9) | 21 (60.0) | 4 (66.7) |
| | Completed high school | 0 | - | 10 (18.2) | 4 (5.7) | 1 (16.7) |
| | Certificate/diploma | 1 (14.3) | - | 6 (10.9) | 2 (5.7) | 1 (16.7) |
| | Undergraduate degree | 4 (57.1) | - | 1 (1.8) | 1 (2.9) | 0 |
| | Postgraduate degree | 2 (28.6) | - | 0 | 0 | 0 |
| **Risk category**[4]: n (%) | In serodiscordant relationship | N/A | N/A | 47 (85.5) | 5 (13.9) | 0 |
| | Migrant worker | N/A | N/A | 1 (1.8) | 0 | 0 |
| | Partner of migrant worker | N/A | N/A | 8 (14.5) | 2 (5.6) | 0 |
| | Multiple partners | N/A | N/A | 3 (5.5) | 4 (11.1) | 0 |
| | Does not trust partner | N/A | N/A | 2 (3.6) | 4 (11.1) | 1 (16.7) |
| | Female sex worker | N/A | N/A | 1 (1.8) | 17 (47.2) | 3 (50.0) |
| | Pregnant/lactating woman | N/A | N/A | 3 (5.5) | 0 | 1 (16.7) |
| | Other | N/A | N/A | 0 | 2 (5.6) | 0 |
| **Total duration on PrEP**[5]: months; mean (sd); range | | N/A | N/A | 8.0 (8.0); 2 days–31 months | 4.1 (4.4); 3 days–24 months | N/A |

[1] Demographic information presented represents characteristics of the 105 healthcare providers who participated in 11 focus group discussions conducted as part of a larger qualitative study. A subset of health providers (n = 51) participated in the pile sorting and ranking exercise.

[2] Information on years in position was missing for one healthcare provider.

[3] Information on educational attainment was missing for one former PrEP user.

[4] Respondents could fall into more than one risk category.

[5] Information on total duration on PrEP was missing for two former and two current PrEP users.

- = information not collected

N/A = not applicable.

Facility-based PrEP promotion, partnership with community leaders, CBOs, and religious leaders were prioritized higher as interventions for women than for men. PrEP promotion in local *shebeens* (bars) was more highly prioritized for men compared to women. Overall, the least prioritized intervention for PrEP uptake was partnership with traditional healers. In head-to-head pairwise elections and when pooling all participants (Table 3), community-based HIV testing emerged as the overall winner for the most helpful intervention to increase PrEP uptake for men and women.

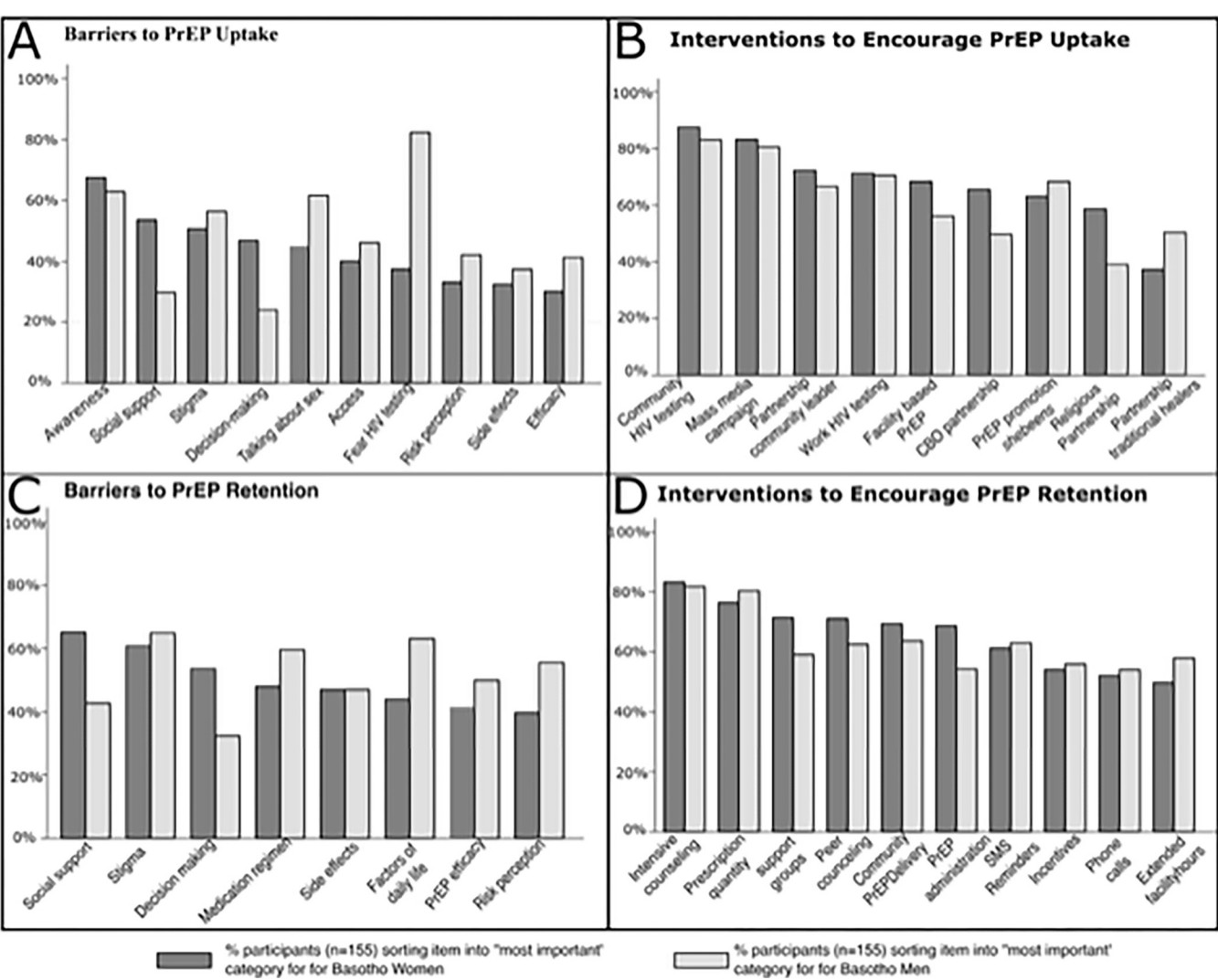

**Fig 2. Participant-prioritized barriers and interventions respectively sorted into the 'most important' and 'most helpful' piles, and stratified by relevance for Basotho women and men.**

### Barriers to PrEP retention

Participant prioritizations indicated that for both men and women in Lesotho, stigma and concern about side effects are important barriers to retention (see Fig 2C). However, other social factors–such as social support and decision-making power–were prioritized more as barriers to PrEP adherence for women than men. Barriers that disrupt daily routines–such as factors of daily life and the daily medication regimen–were prioritized more for men than women. Furthermore, the perception that one is not at risk for HIV infection, and that PrEP is not efficacious, were ranked higher as a barrier for men. In head-to-head pairwise contests and when pooling all participants (Table 3), perceived and/or experienced stigma emerged as the biggest barrier to PrEP retention for both men and women.

### Interventions to increase PrEP retention

To encourage PrEP adherence and retention, participants prioritized intensive counseling and an increase in prescription quantity as the most helpful interventions for both men and

**Table 3. Barriers and intervention candidates ranked as most important / helpful for women and men.**

| | Overall Winner (N = 155) | Policymakers/ Implementing Partners (n = 7) | Health Providers (n = 51) | Current PrEP Users (n = 55) | Former PrEP Users (n = 36) | PrEP Decliners (n = 6) |
|---|---|---|---|---|---|---|
| **Barriers to PrEP Uptake** | | | | | | |
| A1. Limited awareness of PrEP | W | ♀ | ♀ | ♀ | ♀ | ♀♂ |
| A2. Difficulty in communicating with health providers about sexual matters | | | | | | |
| A3. Difficulty in accessing PrEP | | | | | | |
| A4. Fear of HIV testing | M | ♂ | ♂ | ♂ | ♂ | ♂ |
| A5. Perceived stigma | | ♀ | ♀ | | | |
| A.6 Risk perception | | | | | | |
| A.7 Perception that PrEP is not effective | | | | | | |
| A.8 Limited decision making power | | | | | | |
| A.9 Concern of side effects | | | | | | |
| A.10 Lack of social support | | | ♀ | | | |
| **Interventions for PrEP Uptake** | | | | | | |
| B1. Community-based HIV testing | WM | | | ♀♂ | ♀♂ | ♀♂ |
| B2. Workplace HIV testing/PrEP promotion | | | | | | ♂ |
| B3. PrEP promotion in *Shebeens* | | | | | | |
| B4. Facility-based PrEP Promotion | | | | | | |
| B5. Mass media campaign | | ♂ | ♀♂ | | | |
| B6. Partnership with faith-based organizations and religious leaders* | | | | | | |
| B7. Partner with traditional healers* | | | | | | |
| B8. Partner with CBOs* | | | | | | |
| B9. Partner with community leaders* | | ♀ | | | | |
| **Barriers for PrEP Retention** | | | | | | |
| C1. Perceived and/or experienced stigma | WM | ♀♂ | ♀ | ♀♂ | ♀♂ | |
| C2. Risk perception | | | | ♀♂ | ♂ | |
| C3. Perception that PrEP is not effective | | | | | ♀♂ | |
| C4. Decision making power | | ♀ | | ♀ | | |
| C5. Side effects | | | | ♀ | | ♀♂ |
| C6. Medication regimen | M | ♂ | | | | |
| C7. Lack of social support | | | | ♀ | | |
| C8. Factors of daily life | M | ♂ | ♂ | | ♂ | |
| **Interventions for PrEP Retention** | | | | | | |
| D1. Home/community PrEP delivery | | | | ♀ | ♀♂ | |
| D2. Increase PrEP prescription quantity | M | ♀♂ | | ♂ | | ♀♂ |
| D3. SMS reminders | | | | | | ♂ |
| D4. Telephone calls | | | | | | |
| D5. Extended health facility hours | | | | | | |
| D6. Intensive counseling | WM | | ♀♂ | | | |
| D7. PrEP administration | | | | | | |
| D8. Peer counseling | | | | | | |
| D9. Incentives | | | | | | ♂ |

(*Continued*)

**Table 3.** (Continued)

| | Overall Winner (N = 155) | Policymakers/ Implementing Partners (n = 7) | Health Providers (n = 51) | Current PrEP Users (n = 55) | Former PrEP Users (n = 36) | PrEP Decliners (n = 6) |
|---|---|---|---|---|---|---|
| D10. Support groups | | | | | | |

*For HIV testing and PrEP promotion

**W** Overall winning candidate(s) ranked as most important/helpful for women in Lesotho

**M** Overall winning candidate(s) ranked as most important/helpful for men in Lesotho

♀ Winning candidate(s) ranked as most important/helpful for women in Lesotho by participant group

♂ Winning candidate(s) ranked as most important/helpful for men in Lesotho by participant group

women (see Fig 2D). In terms of sex, different modes of PrEP administration, peer counseling and the use of support groups were prioritized as being more helpful interventions for women compared to men. Conversely, extended healthcare facility hours was more highly prioritized for men than women. In head-to-head pairwise elections (Table 3), intensive counseling emerged as the most helpful intervention to increase PrEP retention for women. For men, a Smith set consisting of increase in prescription quantity and intensive counseling were the overall winning candidates.

## Discussion

This study employed participant-centered methodology to identify highly prioritized barriers and interventions for PrEP uptake and retention in Lesotho from various stakeholders. Our findings highlighted gendered differences for barriers to PrEP uptake, with our participants prioritizing low awareness for women and fear of HIV testing for men as the most important barrier. For both women and men, community-based HIV testing was prioritized as the most promising intervention to increase PrEP uptake, with mass media campaigns also ranked highly by participants. Once initiated on PrEP, our participants ranked perceived/experienced stigma as the most important contributor to discontinuation. For men, factors of daily life, such as travel and the inconveniences incurred by the need to take a pill every day, were also ranked highly. As a means to increase retention on PrEP, our participants prioritized intensive adherence counseling and increasing the amount of PrEP dispensed at each visit as interventions that would be most helpful.

Our findings demonstrate convergence across respondent groups–from policymakers to end-users–with regard to barriers that hinder PrEP uptake in Lesotho. In ranking salient barriers for PrEP retention, however, there was divergence; while stigma was highly prioritized across all groups, there were some differences in the ranking of other adherence barriers. Of note, PrEP end-users prioritized factors directly associated with taking PrEP–such as side effects, PrEP efficacy, and perception that they are not at risk–as being salient. These barriers were not prioritized by policymakers and implementing partners. Slight divergences in ranking among our participants are compelling for several reasons. First, they demonstrate the complexity around factors contributing to PrEP retention, illustrating that challenges for adherence are multifaceted, particularly for men. While there has been recent work done on PrEP adherence, few studies have focused exclusively on heterosexual African men [13, 34, 35]. The studies that have, have done so in the context of men in a serodiscordant relationship [36, 37]. Second, this finding is indicative of the varied views held by end-users, healthcare providers, and policymakers. Research shows user concerns emerge from personal and lived experience, whereas implementing partners assess the situation from a birds-eye view [38, 39]. This

gives credence to the importance of incorporating multiple viewpoints whilst developing an intervention–one which is acceptable to the end-user, while simultaneously being feasible from the financial and policy vantage points.

Social factors–including difficulty in discussing sexual matters with health providers, perceived/experienced stigma, and decision-making ability–were prioritized more highly as important barriers to PrEP uptake and retention, particularly for women, than those related to access, efficacy and side effects. Other studies have indicated the importance of considering the social context in the acceptability and adoption of health interventions [15, 40]. In these studies, the primary factors dissuading eligible individuals from initiating or adhering to PrEP were discussed in the context of others conflating PrEP use with being HIV positive, discouragement from others or a moral judgement about the reasons for the individual's PrEP use [17, 41]. To alleviate trepidation arising from concern around stigma, user-centered studies have prioritized packaging and delivery methods, such preference for formulations that promote discrete use and are female initiated. These include the vaginal ring [42], and long-lasting injectables [43, 44].

Fear of HIV testing emerged as a substantial barrier for initiating men on PrEP. Given frequent interactions with the health system–particularly during pregnancy, childbirth and the postpartum period–women have generally been shown to harbor less fear of testing compared to their male counterparts, but are still concerned about disclosing their status [45, 46]. As the first step in the PrEP cascade, HIV testing is essential in determining an individual's eligibility for the drug [29]. In other studies, fear of HIV testing is linked to concern for stigmatization, distress that the result will be positive, and apprehension over confidentiality [40, 47–50]. To encourage HIV testing and address testing-related concerns, interventions that normalize testing and are conducted in the community, or alone (self-tests) have demonstrated higher successes in increasing testing [51, 52].

Perhaps our most relevant finding for the design and implementation of PrEP programs for the general population is the suggestion that interventions be brought closer to people–in this case, PrEP delivery within communities. By bringing PrEP and HIV testing services closer to the community, barriers related to transportation and distance are minimized [50]. Nonetheless, study respondents also prioritized facility-based interventions as useful to encourage adherence, suggesting that a multipronged intervention both at the community and facility level can be employed in tandem. Facility-based interventions could include intensive adherence counseling and monitoring and multi-month PrEP prescriptions [37, 53].

As PrEP is adopted more globally, it is essential to understand end-user preferences while also being cognizant of practicalities policymakers, implementing partners and health providers navigate. While literature proposes interventions to improve PrEP coverage, it is impractical to study the effectiveness of all interventions in resource-limited settings. Therefore, obtaining and ranking the views of stakeholders allows for the most promising interventions to be distilled. This study employed a novel approach to analyze ranked data by applying social choice theory to health. Traditionally, preference data has been gathered qualitatively, with quantitative methods (including discrete choice experiments) being employed in recent years. However, these studies have focused mainly on PrEP formulation preferences [42–44]. We chose to use the Smith voting analysis methods since, unlike political elections, more than one salient candidate is acceptable.

Our study has important limitations. First, due to logistical and sample size constraints, we were unable to conduct this study among a sample of participants that are representative of Lesotho's population. Instead, we employed a purposive sampling strategy with the aim of including a wide range of stakeholders and, thus, views on PrEP delivery. Second, participants were limited by the barrier/intervention candidates presented to them. We sought to mitigate

this limitation by inviting the inclusion of additional barriers or interventions at interview out-set. However, none of the participants suggested additional candidates. Third, our sample size for policymakers, implementing partners, and PrEP decliners was small, making it difficult to generalize findings for these participant groups. We were limited by the number of individuals directly working on the PrEP program in Lesotho when selecting policymakers and imple-menting partners. Additionally, lack of official records for PrEP decliners limited our ability to actively recruit further participants into the study. Lastly, women were overrepresented in our study participants. This may be reflective of earlier iterations of the Lesotho PrEP program that targeted key populations, including serodiscordant couples, female sex workers, and ado-lescent girls and women.

## Conclusion

Our novel participant-centered ranking methodology offered rich insight from varying per-spectives, and particularly from end-users whose opinions are not often considered in the development and implementation of health interventions. The views of this wide range of stakeholders could provide a useful starting point for design and implementation choices of PrEP delivery programs for the general adult population. Views and preferences may, how-ever, vary by setting such that care should be taken in extrapolating any of our findings beyond the Lesotho context.

## Supporting information

**S1 Data. Deidentified dataset for PrEP card sorting and raking for PrEP coverage in Leso-tho.**
(DTA)

**S1 Table. Barrier and intervention candidates presented to participants during the card sorting and ranking exercise.**
(PDF)

**S1 Text. Card sorting and raking instrument for PrEP coverage in Lesotho.**
(PDF)

## Acknowledgments

Our gratitude goes to the many participants from the five Lesotho districts who gave their time, expertise, and experience for this work. We thank the Ministry of Health for their contin-uous support throughout the preparation, data collection, and analysis of this work. We also thank all data collectors for their commitment and diligence.

## Author Contributions

**Conceptualization:** Joy J. Chebet, Shannon A. McMahon, Rachel P. Chase, Tapiwa Tarum-biswa, Chivimbiso Maponga, Esther Mandara, Till Bärnighausen, Pascal Geldsetzer.

**Data curation:** Joy J. Chebet.

**Formal analysis:** Joy J. Chebet, Rachel P. Chase.

**Funding acquisition:** Till Bärnighausen, Pascal Geldsetzer.

**Investigation:** Shannon A. McMahon, Tapiwa Tarumbiswa.

**Methodology:** Joy J. Chebet, Rachel P. Chase, Pascal Geldsetzer.

**Project administration:** Joy J. Chebet, Shannon A. McMahon, Tapiwa Tarumbiswa, Chivimbiso Maponga, Esther Mandara, Till Bärnighausen, Pascal Geldsetzer.

**Supervision:** Joy J. Chebet, Shannon A. McMahon, Chivimbiso Maponga, Esther Mandara, Pascal Geldsetzer.

**Visualization:** Joy J. Chebet.

**Writing – original draft:** Joy J. Chebet, Pascal Geldsetzer.

**Writing – review & editing:** Joy J. Chebet, Shannon A. McMahon, Rachel P. Chase, Tapiwa Tarumbiswa, Chivimbiso Maponga, Esther Mandara, Till Bärnighausen, Pascal Geldsetzer.

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
