## [Decision Letter · Decision Letter 0]

13 Jun 2023

PGPH-D-22-01926

Stakeholder perspectives on interventions to improve HIV Pre-Exposure Prophylaxis uptake and continuation in Lesotho: A participant-ranked preferences study

Dear Dr. Geldsetzer,

Thank you for submitting your manuscript to PLOS Global Public Health. After careful consideration, we feel that it has merit but does not fully meet PLOS Global Public Health’s publication criteria as it currently stands. Therefore, we invite you to submit a revised version of the manuscript that addresses the points raised during the review process.

The review comments can be found at the end of this email, together with any comments from the Editorial Office regarding formatting changes or additional information required to meet the journal's policies at this time. 

Please note that your revision may be subject to further review and that this initial decision does not guarantee acceptance.

We look forward to receiving your revised manuscript.

Kind regards,

Mbuzeleni Hlongwa, Ph.D

Academic Editor

Journal Requirements:

1. We have noticed that you have uploaded Supporting Information files, but you have not included a list of legends. Please add a full list of legends for your Supporting Information files after the references list. 

2. In the online submission form, you indicated that "A deidentified version of our dataset will be made available in a publicly accessible data repository upon acceptance of the manuscript for publication". All PLOS journals now require all data underlying the findings described in their manuscript to be freely available to other researchers, either 1. In a public repository, 2. Within the manuscript itself, or 3. Uploaded as supplementary information.

Additional Editor Comments (if provided):

Reviewers' comments:

Reviewer's Responses to Questions

**Comments to the Author**

1. Does this manuscript meet PLOS Global Public Health’s publication criteria? Is the manuscript technically sound, and do the data support the conclusions? The manuscript must describe methodologically and ethically rigorous research with conclusions that are appropriately drawn based on the data presented.

Reviewer #1: Yes

Reviewer #2: Yes

2. Has the statistical analysis been performed appropriately and rigorously?

Reviewer #1: I don't know

Reviewer #2: N/A

3. Have the authors made all data underlying the findings in their manuscript fully available (please refer to the Data Availability Statement at the start of the manuscript PDF file)?

Reviewer #1: No

Reviewer #2: Yes

4. Is the manuscript presented in an intelligible fashion and written in standard English?

Reviewer #1: Yes

Reviewer #2: Yes

5. Review Comments to the Author

Reviewer #1: Overall:

This study employed a novel staged sorting and ranking method for identifying barriers and interventions to PrEP uptake and persistence among multi-level stakeholders in the context of national PrEP implementation in Lesotho. The methodology employed is novel and makes an important contribution to understanding not only the scope of perceived barriers to PrEP initiation and persistence, but also the relative importance of such barriers and how they vary by gender and among different stakeholders. Ranking methods are advantageous because they force participants to choose relative importance, though they can be unreliable due to the high cognitive load of ordering long lists of items. The authors address this limitation by first asking participants to identify the most important items, thereby shortening the list of items for consideration before ranking. The manuscript makes an important contribution towards the literature, and particularly toward identifying strategies of PrEP delivery that are tailored towards the needs and preferences of different populations, and I think it merits publication once the authors address the issues I outline below. My main suggestion is that the authors provide greater detail and justification for the barriers and interventions chosen, along with proposed mechanisms for how interventions are hypothesized to address the included barriers. Please see my detailed comments below:

Major:

1. The authors do not provide sufficient information on how the barriers and interventions were selected for the survey. Please provide additional theoretical justification for why these particular barriers and interventions were chosen, including a theoretical model if one was used.

2. The study nicely highlights important barriers and important interventions for PrEP uptake and persistence as perceived by multi-level stakeholders. However, I think more could be done to connect identified barriers to interventions of greatest importance. For example, stigma, factors of daily life, and medication regimen were selected as barriers to persistence for men, and multi-month prescriptions and intensive counseling were selected as most important interventions, though it is likely that each of these interventions is addressing different barriers. Use of a theoretical model or mechanism map (#1 above) would help elucidate hypothesized relationships, and an analysis stratified by most important barrier reported would provide further empirical evidence for relationships between barriers and selected interventions. For example, it would be helpful to know if the Smith set of most important interventions differed for individuals who selected stigma vs. factors of daily life as the greatest barrier to PrEP persistence.

3. Can the authors share the materials used to describe the barriers and interventions to participants? Some of the barriers are difficult to interpret without additional detail on how they were presented (e.g. “medication regimen”, “factors of daily life”).

4. My understanding is that gender-stratified results are based on the entire sample reporting importance of barriers and interventions for men and women separately. The authors note that men are under-represented in the study, however I would suggest expanding on this limitation to mention that the results for men are mostly based on perceptions of barriers and importance as reported by women. Alternatively, the authors could consider a sensitivity analysis that was stratified by participant gender (i.e. men reporting on barriers/interventions for men, and women reporting on barriers/interventions for women). If results are similar, this would strengthen the validity of the findings.

5. The sorting/ranking method used is novel and makes an important contribution. In the discussion, I would suggest adding a brief paragraph describing how this method differs from other ranking methods (e.g. best-worst scaling, Q-methodology), including advantages and disadvantages.

Minor:

1. Intervention, Line 108: the references for PrEP implementation studies in sub-Saharan Africa are quite dated (2016) and there are more recent studies published in this area. Please update references and place this study into context with more recent studies.

2. In table 2, the demographics are presented for 105 healthcare workers who participated in a larger study, but only 51 participated in the present study. Please include demographic data specific to the 51 participants in the present study.

3. In table 3, I think there is an error in the ‘Policy makers/implementing partners’ column for Barriers to PrEP uptake. Should limited awareness of PreP and perceived stigma be indicated for women instead of men?

Reviewer #2: In this manuscript, the authors present descriptive findings from a ranked preferences exercise among a range of HIV pre-exposure prophylaxis (PrEP) implementation stakeholders (including program staff, healthcare workers and end users of PrEP), designed to understand perceptions of barriers and potential interventions to: 1) PrEP uptake and 2) retention in PrEP programs, among men and women in Lesotho. Overall, the manuscript is well written and easy to follow. The results are descriptive without statistical analyses. Novel aspects of the study include consideration of stakeholder perceptions as ranked preferences determined with a card sorting exercise and inclusion of persons who are currently on PrEP, no longer on PrEP but PrEP experienced, and those who declined PrEP. The key weaknesses of the manuscript are as follows.

First, the authors note in the Introduction that, “At present, little evidence on interventions to achieve high PrEP coverage in the general population is available, particularly in sub-Saharan Africa.” The authors have overlooked several publications from the SEARCH study in East Africa in which PrEP has been offered to a general population (for example, please see Koss et al, “Uptake, engagement, and adherence to pre-exposure prophylaxis offered after population HIV testing in rural Kenya and Uganda: 72-week interim analysis of observational data from the SEARCH study,” Lancet HIV, 2020, PMID: 32087152), as well as the Partners Demonstration project that incorporated PrEP into HIV care clinics (reaching both men and women, see reference: Irungu et al, "Integration of pre-exposure prophylaxis services into public HIV care clinics in Kenya: a pragmatic stepped-wedge randomised trial," Lancet Global Health, 2021, PMID: 34798031).

Second, the authors do not justify why they used the card sorting exercise to determine ranked preferences as opposed to other quantitative preference elicitation methods, such as best worst sampling surveys, or discrete choice experiments. The authors should provide a brief explanation at how they arrived at this method. For an overview of types of preference elicitation methods, see Soekhai et al, “Methods for exploring and eliciting patient preferences in the medical product lifecycle: a literature review,” Drug Discov Today, 2019, PMID: 31077814; as well as Kerkhoff et al, “A world of choices: preference elicitation methods for improving the delivery and uptake of HIV prevention and treatment,” Curr Opin HIV AIDS. 2023, PMID: 36409315.

Third, the authors recently published findings from qualitative interviews with (what appears to be) the same population of stakeholders in PLoS Global Public Health (reference: Geldsetzer et al, “Knowledge and attitudes about HIV pre-exposure prophylaxis: Evidence from in-depth interviews and focus group discussions with policy makers, healthcare providers, and end-users in Lesotho,” 2022, PMID: 36962565). A weakness of the present manuscript (under review) is a lack of discussion of what the card sorting exercise adds to the prior publication, and it seems unusual that the authors have not cited this manuscript as a useful reference for readers to get a better picture of the viewpoints of this group of stakeholders in Lesotho. Did the qualitative interviews and analysis inform the card sorting exercise, or vice versa?

6. PLOS authors have the option to publish the peer review history of their article (what does this mean?). If published, this will include your full peer review and any attached files.

**Do you want your identity to be public for this peer review?** For information about this choice, including consent withdrawal, please see our Privacy Policy.

Reviewer #1: No

Reviewer #2: No

<quillbot-extension-portal></quillbot-extension-portal>

---

## [Decision Letter · Decision Letter 1]

5 Sep 2023

Stakeholder perspectives on interventions to improve HIV pre-exposure prophylaxis uptake and continuation in Lesotho: A participant-ranked preferences study

PGPH-D-22-01926R1

Dear Dr. Geldsetzer,

We are pleased to inform you that your manuscript 'Stakeholder perspectives on interventions to improve HIV pre-exposure prophylaxis uptake and continuation in Lesotho: A participant-ranked preferences study' has been provisionally accepted for publication in PLOS Global Public Health.

Best regards,

Mbuzeleni Hlongwa, Ph.D

Academic Editor

Reviewer Comments (if any, and for reference):

Reviewer's Responses to Questions

**Comments to the Author**

1. If the authors have adequately addressed your comments raised in a previous round of review and you feel that this manuscript is now acceptable for publication, you may indicate that here to bypass the “Comments to the Author” section, enter your conflict of interest statement in the “Confidential to Editor” section, and submit your "Accept" recommendation.

Reviewer #1: All comments have been addressed

Reviewer #3: All comments have been addressed

2. Does this manuscript meet PLOS Global Public Health’s publication criteria? Is the manuscript technically sound, and do the data support the conclusions? The manuscript must describe methodologically and ethically rigorous research with conclusions that are appropriately drawn based on the data presented.

Reviewer #1: Yes

Reviewer #3: Yes

3. Has the statistical analysis been performed appropriately and rigorously?

Reviewer #1: N/A

Reviewer #3: N/A

4. Have the authors made all data underlying the findings in their manuscript fully available (please refer to the Data Availability Statement at the start of the manuscript PDF file)?

Reviewer #1: Yes

Reviewer #3: Yes

5. Is the manuscript presented in an intelligible fashion and written in standard English?

Reviewer #1: Yes

Reviewer #3: Yes

6. Review Comments to the Author

Reviewer #1: (No Response)

Reviewer #3: I read this article with great interest, as I was unfamiliar with the methods used, but interested in the core questions asked. I can see that the authors have addressed the points raised by previous reviewers sufficiently and that there is clarity around both the use of methods and the findings. I was struck by some of the differences between health workers/policy makers and PrEP users and glad to see the differences raised in your discussion. I think this article will make an important contribution to addressing the gendered differences in PrEP provision and support and to better focus on how to make the most of limited resources when rolling out and/or improving PrEP programmes.

7. PLOS authors have the option to publish the peer review history of their article (what does this mean?). If published, this will include your full peer review and any attached files.

**Do you want your identity to be public for this peer review?** For information about this choice, including consent withdrawal, please see our Privacy Policy.

Reviewer #1: No

Reviewer #3: No
